# Importance of Footwear Outsole Rigidity in Improving Spatiotemporal Parameters in Patients with Diabetes and Previous Forefoot Ulcerations

**DOI:** 10.3390/jcm9040907

**Published:** 2020-03-25

**Authors:** Mateo López-Moral, Raúl Juan Molines-Barroso, Francisco Javier Álvaro-Afonso, Luigi Uccioli, Eric Senneville, José Luis Lázaro-Martínez

**Affiliations:** 1Diabetic Foot Unit, Facultad de Medicina, Universidad Complutense de Madrid, Instituto de Investigación Sanitaria del Hospital Clínico San Carlos (IdISSC), 28040 Madrid, Spain; matlopezmor@hotmail.com (M.L.-M.); alvaro@ucm.es (F.J.Á.-A.); diabetes@ucm.es (J.L.L.-M.); 2Diabetic Foot Unit, Department Systems Medicine, University of Rome Tor Vergata, 00133 Rome, Italy; luccioli@yahoo.com; 3Infectious Diseases Department, Gustave Dron Hospital, 59200 Tourcoing, France; esenneville@ch-tourcoing.fr

**Keywords:** rigid rocker sole, diabetic polyneuropathy, ulcer prevention, spatiotemporal parameters

## Abstract

We aimed to identify if any differences existed in spatiotemporal parameters during gait among different densities of rocker soles in patients with a history of neuropathic ulcerations and the differences in comfort between shoe conditions. This study was a cross-sectional study of 24 patients with diabetes and a history of neuropathic diabetic foot ulcers (DFUs). Spatiotemporal parameters (duration of stance phase (ms), stride length (cm), and step velocity (m/s)) were analyzed in barefoot, semirigid outsole, and rigid outsole footwear conditions. A dynamic pressure measurement system (Footscan^®^ system, RSscan International, Olen, Belgium) was used to assess shoe conditions. We also analyzed differences in comfort between the shoe conditions using a visual analog scale. A Wilcoxon test for paired samples was used to assess gait differences. Result showed that a rigid outsole causes changes in the subphases of the stance phase (*p* < 0.001; Cohen d = 0.6) compared to a semirigid outsole. Stride length (*p* < 0.001; Cohen d = 0.66) and step velocity were significantly longer (*p* < 0.001; Cohen d = 2.03) with the use of rigid outsole footwear. A rigid rocker sole reduces the time of the stance phase, in addition to increasing the stride length and velocity of step in patients with a previous history of DFUs.

## 1. Introduction

Diabetic foot ulcers (DFUs) are one of the most common complication of diabetes mellitus [1], accounting for 20% of minor and major amputations due to DFU-induced infections [2]. Patients with an active DFU have a 2.5 times higher risk of death [3], and the mortality rate increases to 70% 5 years after undergoing an amputation [4]. In 2017, DFUs increased the treatment cost of diabetes to $176 billion in the US, making it more expensive to treat than common cancer complications [5].

Almost 50% of DFUs appear on the plantar surface of the forefoot [6], and diabetic polyneuropathy (DPN) plays a key role in their development, due to reduced proprioception of the patients and reduced range of motion of foot joints [7]. The combination of DPN, a history of DFUs, and minor amputations significantly increases the risk of reulceration to 40% in the first year after ulcer healing [1].

Appropriate metabolic control, vascular sufficiency, self-monitoring, padding, medical grade orthoses (depending on need and foot type), and therapeutic footwear for high risk patients are part of the main strategies for preventing diabetic foot disease and its complications [8]. It has been demonstrated that using therapeutic footwear reduces the risk of ulcer recurrence, specifically, recurrent ulcers on the plantar surface of the metatarsal heads [9]. International guidelines on diabetic foot care are currently focused on the prevention of DFUs, for this reason, multiple interventions have been developed to prevent diabetic foot complications [10,11]. A rigid rocker sole effectively reduces the rate of plantar reulceration [12]. 

Previous studies have shown that a rigid rocker sole decreases plantar pressure in the forefoot [13] and changes spatiotemporal parameters in comparison with a flattened sole. Moreover, it has been shown that rigid materials result in kinematic parameter changes when compared to semirigid materials in healthy patients without an altered gait [14]. Patients with DPN have shown longer midstance phases in the gait cycle [15,16]. Furthermore, some studies have shown that patients with DPN have decreased spatiotemporal parameters secondary to muscle dysfunction and decreased range of motion, such as stride length and step velocity during gait cycle, when compared to controls. This can lead to a change in the joint kinematics and muscle activation patterns, and thus could increase the risk of foot complications such as elevated plantar pressure [16]. 

Altered spatiotemporal parameters, such us stride length, speed step, and midstance phase duration have shown changes in plantar pressure patterns, and thus increase the risk of ulcer occurrence and further foot complications [16]. We still do not know how the spatiotemporal parameters of patients with neuropathy and previous DFUs are affected by rocker sole density. Therefore, this study primarily aimed to analyze the differences in spatiotemporal parameters during gait between different densities of rocker soles in patients with a history of neuropathic ulceration. We also aimed to analyze the differences in comfort between the shoe conditions. We hypothesized that a rigid rocker sole positively benefits kinematic parameters in patients with diabetic foot complications. 

## 2. Methods

### 2.1. Subjects

This cross-sectional study recruited 24 patients with diabetes and a history of neuropathic DFUs on the forefoot (metatarsal heads and interphalangeal joint of the hallux) from the Diabetic Foot Unit of the Complutense University of Madrid, Spain. The target sample size was calculated using Epidat^®^ version 4.2 for Mac OS (Consellería de Sanidade, Xunta de Galicia, 15703, España; Organización Panamericana de la Salud (OPS-OMS); Universidad CES, Medellín, Antioquia, Colombia). It was determined that the standard deviation was 10 ms to detect a difference of at least 6 ms for stance phase duration as primary outcome [15], based on a desired power of 80% with a β level of 20%, α level of 0.05, and confidence interval of 95%. Assuming a loss of 0% due to the cross-sectional study design, at least 24 participants were included in the study.

The inclusion criteria were confirmed type 1 or type 2 diabetes, age > 18 years, loss of protective foot sensation because of peripheral neuropathy, and previous foot ulcers under the metatarsal head or interphalangeal joint of the hallux in at least one foot. The exclusion criteria were ulcers during examination, transmetatarsal or major amputation (below or above the knee), history of rheumatoid disease, other causes of neuropathy, critical limb ischemia as defined according to the International Working Group on Diabetic Foot guideline [17], Charcot foot, foot deformity that does not allow standard therapeutic footwear, and the need for walking aids. 

This study was approved by the local ethics committee of the Hospital Clínico San Carlos, (Madrid, Spain) by an amendment on June 2018 (Code: 16/408–P). Patients were included only after providing informed consent.

### 2.2. Clinical Evaluation

DPN was diagnosed according to the inability to sense the pressure of a 10-g Semmes–Weinstein monofilament at three plantar foot sites and/or a vibration perception threshold > 25 V as assessed using a biothesiometer (Me.Te.Da. s.r.l., Via Silvio Pellico, 4, 63074 San Benedetto del Tronto, Italy) [10,18]. Critical limb ischemia was diagnosed as an absence of both distal pulses and an ankle brachial index of < 0.39, ankle systolic pressure < 50 mmHg, and toe pressure < 30 mmHg or transcutaneous oxygen pressure < 30 mmHg [17]. Body mass index was calculated as weight (kg) divided by height (m^2^). The clinicopathologic data collected included diabetes type, mean duration of diabetes, and glycated hemoglobin (HbA1c) values in the previous 3 months. The patients’ renal and retinopathy statuses were assessed. Forefoot deformities were defined as the presence of at least one of the following conditions: hallux valgus; Taylor’s bunion; metatarsal head bone prominences; or toe contractures such as hammertoe, claw-toe, or mallet-toe deformities [19,20]. Foot position was stratified into neutral (+1 to +7), pronated (> +7) and supinated (< 0), according to the Foot Posture Index (FPI-6) [21].

### 2.3. Shoe Conditions 

Patients were consecutively recruited based on the shoe size for both rigid and semirigid outsole shoes (40, 41, 42, and 43 for men and 37, 38, 39, and 40 for women). Patients wore two different therapeutic footwear (rigid and semirigid rocker outsole, Podartis s.r.l Unipersonale—Crocceta del Montello, 31035 Treviso, Italy) with the same general characteristics: high toe box, enough width to accommodate toe deformities such as claw or hammer toes, wide heel, and buckles for fasteners. Furthermore, the shoes were free of seams, folds, and hollows [8]. All shoes had a rocker sole (i.e., an anteroposterior rocker) for plantar pressure reduction of the metatarsophalangeal joints. The pivot point needed to be proximal to these joints [8]. The rocker angle was defined as the 20 Ŷ angle between the floor and sole under the metatarsal heads [12]. The sole density was categorized into the semirigid rocker sole (Wellwalk technology with Vibram strips, density 240 kg/m^2^, shore A 50) and the rigid (composite fiber, density 330 kg/m^2^, shore A) rocker sole (Figure 1). To ensure the proper position of the pivot point in the rocker outsole, the principal investigator used a weight-bearing lateral X-ray of the shod feet as previously described [12].

### 2.4. Experimental Procedures

#### 2.4.1. Kinematic Analyses

A dynamic pressure measurement system (Footscan^®^ system, RSscan International, 3583 Olen, Belgium) was used to record the maximal mean pressure (kPa) and force time integral (kPa/s) in barefoot conditions. In addition, the following spatiotemporal parameters were checked in three different conditions (i.e., barefoot, semirigid sole footwear, and rigid sole footwear): the stance phase of the gait cycle, stride length, and gait velocity. The hardware included a 2-m plate with four sensors/cm^2^ and a 3D-Box interface that was synchronized with a motion capture system. All data were recorded at a measurement frequency of 500 Hz and were processed using Scientific Footscan^®^ software (RSscan International, 3583 Olen, Belgium).

The stance phase was divided into four subphases (ms) according to the Footscan software results as follows. The first was the initial contact subphase (ICP) (i.e., the period from first foot contact until first metatarsal contact. The second was the forefoot contact subphase (FFCP) (i.e., the period immediately following ICP, until the all metatarsal head areas make contact with the pressure plate). The third was the foot flat subphase (FFP) (i.e., the phase following FFCP and ends when the heel is off the ground)- And the fourth was the forefoot push-off subphase (FFPOP) (i.e., starts when the heel is off the floor and ends when the foot is off the ground) [22].

#### 2.4.2. Comfort Assessment 

Patients were asked to walk barefoot for 3 min in the lab to assess their autonomous gait. Each participant scored the shoe comfort after the last trial und each condition by a novel score system for patients with DPN, by placing a vertical line on a 100 mm visual analog scale (VAS) [13]. The outmost left (0 mm) indicated very uncomfortable and the outmost right (100 mm) indicated very comfortable. The therapeutic footwear trial was conducted in random, using a random number table in order to elect the hardness of the outsole, and thus the patients were blinded to the type of outsole used during gait, to avoid bias in the interpretation of the shoe comfort as both types of footwear had the same appearance on the upper surface.

#### 2.4.3. Data Analyses

Every foot was analyzed individually due to the differences in foot type (FPI), previous ulceration, minor amputation, and foot deformities. Stance phase (ms) was assessed barefoot and in both shoe conditions using a rigid and semirigid rocker sole. Three screenshots were taken, and the mean was calculated for all four subphases. The investigator who analyzed and extracted data from spatiotemporal parameters was blinded to the type of outsole used during trials. Stride length (cm) and gait velocity (m/s) were recorded in the three conditions, and the same protocol was used to calculate the means.

### 2.5. Main Outcome

The main outcome measure was spatiotemporal parameters assessed as follows: time in milliseconds for all four subphases of the stance phase. 

### 2.6. Secondary Outcomes

The secondary outcome measure was stride length (cm) and velocity (m/s) during gait in different shoe conditions and differences in comfort between shoe conditions assessed using a VAS.

### 2.7. Statistical Analyses

Quantitative variables were presented as the median and interquartile range (IQR), while qualitative variables were presented as the percentage and frequencies. A Wilcoxon test for paired samples was used to explore the differences in stance subphases, stride length, velocity during gait, and comfort between different shoe conditions. The Kruskal-Wallis test was used to understand differences between foot type and changes in spatiotemporal parameters during gait. The strength of difference in the effect size was calculated using a phi coefficient for a chi-square test and an r coefficient for a non-parametric test considering the values > 0.01 as a small effect, > 0.30 as a medium effect, and > 0.50 as a large effect. Cohen’s d was calculated as the effect size for the parametric test using an effect size calculator (http://www.uccs.edu/~lbecker/) and considering the values > 0.2, > 0.5, and > 0.8 as small, moderate, and large effects, respectively [23]. All statistical analyses were performed using SPSS statistics version 25.0 for Mac OS (SPSS, Chicago, IL, USA). Furthermore, GraphPad^®^ for Mac OS was used to generate graphics to assess the differences between different shoe conditions and barefoot. *P* values < 0.05 were considered statistically significant with confidence intervals of 95%.

## 3. Results

Baseline data on demographic characteristics, diabetes, and foot complications are shown in Table 1.

Details of foot types and previous amputations to understand the differences between the feet of each patient are shown in Table 2.

The rigid sole condition showed a shorter stance phase time compared to the semirigid sole condition (837 ((IQR); 887.5–805.2) ms vs 957 (1107–840.2) ms; *p* = 0.001; Cohen d = 0.7) and barefoot condition (837 (887.5–805.2) ms vs 918 (1027.8–829) ms; *p* < 0.001; Cohen d = 0.91).

Figure 2 shows the differences (ms) for each stance subphase during gait under different shoe conditions (barefoot, semirigid, and rigid rocker sole).

With respect to the initial contact subphase, it was shorter in the barefoot condition than in the semirigid sole condition (71.5 (106–40) ms vs 127 (146–106) ms; *p* < 0.001; Cohen’s d = 0.22) and rigid sole condition (71.5 (106–40) ms vs 135 (181.8–111) ms; *p* < 0.001; Cohen’s d = 0.26). The initial contact subphase was also shorter in the semirigid outsole condition than in the rigid sole condition (127 (146–106) ms vs 135 (181.8–111) ms; *p* = 0.015; Cohen’s d = 0.14) (Figure 2A). In addition, the forefoot contact subphase was shorter in the barefoot condition than in the rigid sole condition (210.5 (106–40) ms vs 282 (396.8–176.2) ms; *p* = 0.021; Cohen’s d = 0.6). It was also shorter in the semirigid outsole condition compared to the rigid condition (155 (305.2–88.5) ms vs 282 (396.8–176.2) ms; *p* = 0.011; Cohen’s d = 0.22) (Figure 2B). 

Barefoot condition results were shorter compared to that of the semirigid sole condition in the foot flat subphase (406 (517.8–341.2) ms vs 381.5 (436.5–269.8) ms; *p* = 0.038; Cohen’s d = 0.04). The foot flat subphase was shorter in the rigid sole condition than in the barefoot condition (116 (186.5–59.2) ms vs 406 (517.8–341.2) ms; *p* < 0.001; Cohen’s d = 0.82). It was also shorter in the rigid sole condition than in the semirigid condition (116 (186.5–59.2) ms vs 381.5 (436.5–269.8) ms; *p* < 0.001; Cohen’s d = 0.6) (Figure 2C). Finally, in the forefoot push-off subphase, we found significant differences between the barefoot condition and semirigid sole condition (242 (286–182.5) ms vs 257.5 (303–214) ms; *p* = 0.001; Cohen’s d = 0.2). The forefoot push-off subphase was shorter in the rigid outsole condition than in the semirigid condition (271 (333.2–241) ms vs 257.5 (303–214) ms; *p* = 0.012; Cohen’s d = 0.06) (Figure 2D).

With respect to stride length, it was shorter in the barefoot condition than in the semirigid sole condition (39.7 (48.6–30.8) cm vs 46.1 (53.6–35.2) cm; *p* < 0.001; Cohen’s d = 0.35) and rigid condition (39.7 (48.6–30.8) cm vs 52.6 (58.1–45.4) cm; *p* < 0.001; Cohen’s d = 0.67) (Figure 3). For step velocity during gait, differences between shoe conditions are shown in Figure 3. The step velocity was lower in the barefoot condition than in the semirigid sole condition (0.8 (1–0.7) m/s vs 1 (1.2–0.9); *p* < 0.001; Cohen d = 0.95) and the rigid sole condition (0.8 (1–0.7) m/s vs 1.4 (1.6–1.1) m/s; *p* < 0.001; Cohen’s d = 2.03). Meanwhile, it was faster in the rigid sole condition than in the semirigid condition (1.4 (1.6–1.1) m/s vs 1 (1.2–0.9) m/s; *p* < 0.001; Cohen’s d = 1.41).

The stride length was also longer in the rigid outsole footwear than in the semirigid footwear (*p* < 0.001; Cohen’s d = 0.66). Stride length was longer in patients with a supinated foot type and it was shorter in patients with a pronated foot type in comparison with a neutral foot under both shoe conditions (Table 3). Stance subphases and step velocity did not show any difference between different foot types and shoe conditions.

Finally, with respect to comfort, VAS scores were higher in the rigid outsole condition (89 (94–81) mm) than in the semirigid sole condition (62.5 (75–49.2) mm) (*p* < 0.001; Cohen d = 1.38).

## 4. Discussion

The results of our study demonstrated that outsoles that are more rigid changed timing during the step cycle, reducing the final subphases (foot flat and push-off subphases) and lengthening the other subphases, reducing the totality of the stance phase. Moreover, stride length and step velocity increased with the use of more rigid materials in the sole during gait. Finally, patients felt more comfortable with rigid soles than semirigid soles.

It has been hypothesized that a rigid rocker sole restricts the dorsiflexion of the toes, quickening the transition from the total contact phase of the foot to the push-off phase during gait in patients with DM and DPN [16]. Our results support this hypothesis. Metatarsophalangeal joints (MPJ) are necessary during the last subphases of the gait, and those parts of the step cycle were reduced at the cost of increasing the initial subphases. We have proven that footwear outsoles with more rigid materials lead to a shorter time during the push-off and foot flat phases, which could be related to more restriction of dorsiflexion of the MPJ. 

The changes in spatiotemporal parameters with the use of denser materials found in the current study may justify our previous findings [12] and could help explain why harder outsoles might reduce the risk of reulceration among patients with a history of DPN and DFU. According to Reints et al. [13], who observed a reduction in metatarsal plantar pressure and an increase in heel plantar pressure in a non-diabetic population, we have hypothesized that the reduction in the timing of the final subphases at the expense of the increase in the initial subphases could be linked and might explain the reduction in reulceration risk. However, the relationship between plantar pressure and spatiotemporal parameters was not evaluated in the present study, thus, we cannot identify which specific forefoot plantar areas will benefit most from plantar pressure reduction.

We also observed that stride length and gait velocity increased with the use of rigid rocker soles. Stride length even changed in patients with supinated feet. It has been suggested that patients with diabetic foot complications adopt gait patterns with reduced stride length and step velocity [24]. With rigid rocker soles, this altered pattern could be “normalized.” However, we neither know how long these changes in spatiotemporal parameters could last, nor how they affect plantar pressure. Further research is needed to identify which of these spatiotemporal changes observed in the current study resulted in the major plantar pressure decrease.

Patients with a history of neuropathic ulceration have altered gait patterns, increasing plantar pressure in forefoot areas compared to patients with no previous history of DFU or DPN. Raspovic et al. [25] showed that mean values of stride length and step velocity in patients with DPN and previous DFU were 1.2 ± 0.2 m and 1.1 ± 0.2 m/s, respectively. Our results showed that in our population, those with previous amputations and foot deformities had more altered gait patterns, due to the low stride length (0.4 (0.48–0.31) m) and step velocity (0.8 (1–0.7) m/s) in the barefoot condition. Our patients exhibited an increase in step velocity (1.4 (1.6–1.1) m/s) and stride length (0.53 (0.59–0.45) m) derived from the use of harder rocker soles, reaching similar values to those observed in persons with diabetes and no previous history of DFU or DPN [25].

Some studies have found that an increase of step velocity and stride length in healthy subjects was related to an increase of plantar pressure patterns in jogging and running conditions [26]. Thus, the speed increase should suggest an increased risk of ulceration. However, the increase of spatiotemporal parameters found in the current study, under harder outsole conditions, was achieved in walking conditions. Additionally, it was observed in previous research [25] that a moderate increase of step velocity (1.5 km/h) and stride length does not modify plantar pressure patterns during normal gait. Thus, we believe that adjutant therapy with rigid rocker soles will increase stride length and step velocity, then, patients will benefit from an autonomous gait, and reduction of potential risks. Meanwhile, Karimi et al. [27] found no differences in spatiotemporal parameters between the uses of different rocker sole designs among the healthy and subjects with diabetes. This shows that compared with rocker designs, hardness of materials has more impact on the kinematics of gait.

Lin et al. have demonstrated that harder rocker soles decrease the duration of plantar forces during the totality of the stance phase in non-diabetics [14], which was associated with plantar pressure reduction. However, they did not investigate the timing during the step cycle subphases, which were most reduced with the use of harder outsoles. Our results showed that rigid rocker soles are similarly effective in the reduction of the totality of stance phase in patients with diabetic foot complications. 

In addition, our results showed that in spite of a contradictory assumption, a rigid rocker sole is more comfortable. Shoe upper surfaces differed a little between rigid and semirigid outsole footwear (Figure 1), which could have affected the study blindness in some patients. Comfort sensation is a subjective feeling influenced by several factors, including appearance. It has been demonstrated that uncomfortable shoes are among the reasons for the low adherence of patients with diabetes to therapeutic footwear [28], decreasing the efficacy of this preventive treatment [29]. Consequently, prescribing a more comfortable therapeutic footwear could help improve adherence. 

To the best of our knowledge, this is the first study to investigate the differences in spatiotemporal parameters during gait in a population of patients with diabetes, DPN, and previous DFU in the plantar forefoot area. However, our results should also be interpreted with caution because we only studied unworn therapeutic shoes, and the mechanical properties of shoe materials could be impaired with the duration of usage [9]. Further, the duration of the phases of the gait could be different. 

## 5. Conclusions

Compared with a semirigid sole footwear, rigid rocker soles reduce the final subphases of the stance phase and lengthen the initial subphases during gait in patients with a history of neuropathic ulcerations. The findings of this study can help explain the role of different outsole densities in the prevention of reulceration in patients with previous DFUs and DPN.

## Figures and Tables

**Figure 1 jcm-09-00907-f001:**
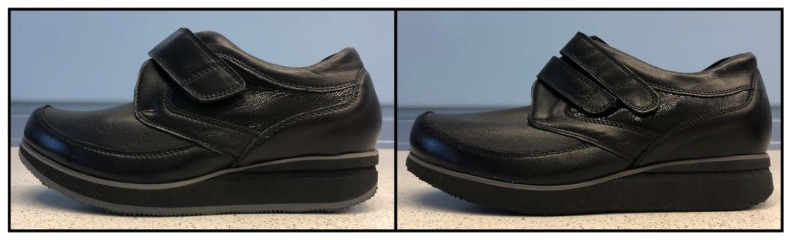
Different shoe types analyzed in the study. Left: semirigid rocker sole footwear; right: rigid rocker sole footwear (Color).

**Figure 2 jcm-09-00907-f002:**
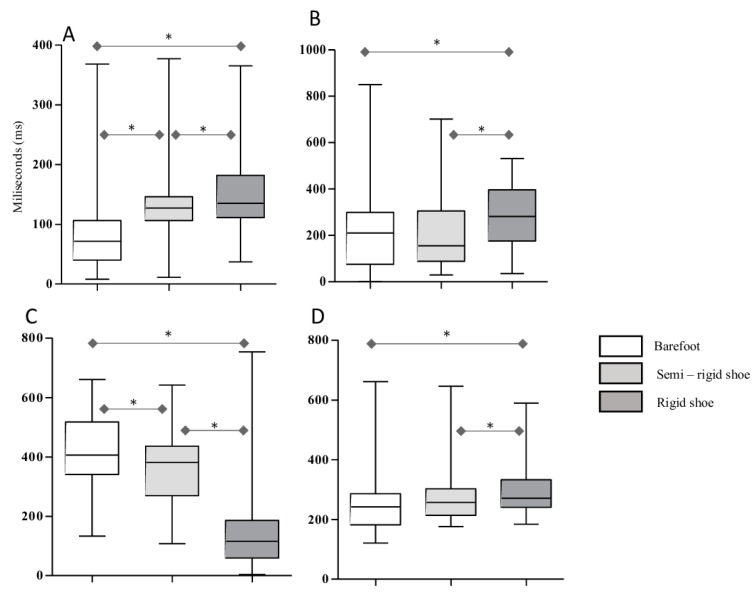
Differences in time (ms) for each stance subphase during gait under different shoe conditions. **A**, Initial contact subphase (ICP). **B**, Forefoot contact subphases (FFCP). **C**, Foot flat subphase (FFP). **D**, Forefoot push-off subphase (FPOP). * *p* < 0.05 indicates statistical significance.

**Figure 3 jcm-09-00907-f003:**
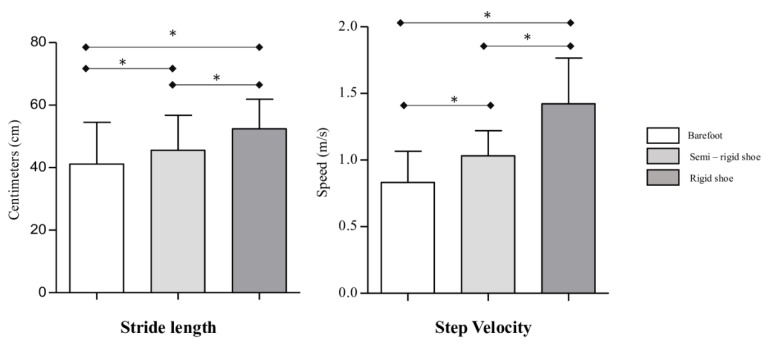
Differences in stride length (cm) and step velocity (m/s) between each study condition. * *p* < 0.05 indicates statistical significance.

**Table 1 jcm-09-00907-t001:** Patients baseline characteristics (*n* = 24) and foot characteristics (*n* = 48).

Baseline Characteristics	Patients (*n* = 24)
Male n (%)	17 (70.8)
Female n (%)	7 (29.2)
Type 1 DM n (%)	3 (12.5)
Type 2 DM n (%)	21 (87.5)
Retinopathy n (%)	15 (62.5)
Nephropathy n (%)	1 (4.2)
Foot deformity n (%)	22 (85)
Diabetic polyneuropathy (%)	24 (100)
Mean age ± SD (years)	62.5 ± 8.8
Body mass index (kg/cm^2^), mean ± SD	27.3 ± 4.6
Glycated hemoglobin mmol/mol (%), mean ± SD	53 (7.03) ± 1.15
Diabetes mellitus (years), mean ± SD	21.3 ± 14.1
Foot Characteristics	*n* = 48 feet
Hallux abductus valgus n (%)	6 (12.5)
Hammer toes n (%)	37 (77.1)
Taylor bunion n (%)	14 (29.2)
Previous amputation n (%)	28 (58.3)
Previous DFU n (%)	39 (81.3)
Interphalangeal joint of the hallux n (%)	2 (5.2)
First MTH n (%)	13 (33.3)
Second MTH n (%)	11 (28.2)
Fourth MTH n (%)	8 (20.5)
Fifth MTH n (%)	5 (12.8)
Foot Posture Index (FPI), mean ± SD	1.6 ± 4.8
Neutral n (%)	26 (54.2)
Pronated n (%)	6 (12.5)
Supinated n (%)	16 (33.3)
Barefoot maximal mean pressure in the forefoot (kPa), mean ± SD	757.2 ± 357.8
Barefoot force time integral in the forefoot (kPa/s), mean ± SD	244.52 ± 156.65

Abbreviations: DM, diabetes mellitus; PAD, peripheral arterial disease; SD, standard deviation; DFU, diabetic foot ulcer; MTH, metatarsal head; Foot Posture Index: pronated feet include values between +1 to +7, pronated > +7 and supinated < 0 according to the FPI-6.

**Table 2 jcm-09-00907-t002:** Foot characteristics (N = 48) of patients by Left (*n* = 24) and Right (*n* = 24) foot.

*n* = 24 Patients	Left Foot *n* = 24		Right Foot *n* = 24	
	Previous Amputation	Foot Posture Index	Previous Amputation	Foot Posture Index
Patient 01	3rd MTH	−3	-	−9
Patient 02	-	2	5th MTH	0
Patient 03	2nd MTH	0	-	0
Patient 04	-	6	-	2
Patient 05	-	5	1st MTH	6
Patient 06	2nd MTH	7	-	3
Patient 07	1st MTH	10	4th MTH	8
Patient 08	4th MTH	9	-	4
Patient 09	-	3	-	2
Patient 10	2nd and 4th MTH	−2	-	−3
Patient 11	-	−5	-	−2
Patient 12	3rd MTH	5	1st MTH	8
Patient 13	-	0	-	−4
Patient 14	-	−1	-	−1
Patient 15	-	2	1st MTH	6
Patient 16	2nd MTH	−2	-	−2
Patient 17	1st MTH	3	5th MTH	−6
Patient 18	5th MTH	5	2nd MTH	0
Patient 19	2nd MTH	10	5th MTH	7
Patient 20	1st MTH	4	-	−2
Patient 21	1st MTH	4	3rd MTH	6
Patient 22	2nd MTH	5	2nd toe	0
Patient 23	1st MTH	4	-	−2
Patient 24	5th MTH	−2	3rd MTH	−12

Abbreviations: MTH, metatarsal head. Foot Posture Index: pronated feet include values between +1 to +7, pronated > +7 and supinated < 0 according to the FPI-6.

**Table 3 jcm-09-00907-t003:** Differences between foot type and stride length under different shoe conditions.

Foot Type				
Stride Length (cm)	Neutral	Pronated	Supinated	*P*-Value
Semirigid condition	21.7	11.42	34.31	0.001 *
Rigid condition	21.71	14.33	32.84	0.007 *

Abbreviations: * *p* < 0.5 indicates significant association.

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
