# Peer review of "Importance of Footwear Outsole Rigidity in Improving Spatiotemporal Parameters in Patients with Diabetes and Previous Forefoot Ulcerations"

_jcm, 2020, doi:10.3390/jcm9040907_

Round 1
Reviewer 1 Report
López – Moral M et al. presented well-designed study about footwear for diabetic patients.
Footwear characteristics are obviously important for the gait cycle and foot health, and their clear data might be helpful for the doctors and staffs for diabetic care, and for the specialists for these shoes.
Some queries are raised.
In the comfort assessment, authors stated the type of outsole were blinded, however, blindness might not be certified because it will be easy for some patients to feel their outsole rigidity. Moreover, shoe upper surfaces look a little bit different between rigid and semi-rigid in Fig.1.
In description of shoe conditions, laces are not seen in Fig.1. Strictly speaking, laces and buckles are different for the ankle fixation and thus will affect the parameters the authors described. Type of shoes should be intimately described if there are some significant differences about parts.
There seems to be big differences in foot type. Therefore, foot type variability might be correlated with the result. Sample size is limited but reference to the effective foot type for the rigid sole is helpful to the readers.
Typo error
P3 disfunction->dysfunction
Author Response
Thank you for accepting the review of the current article and for the suggestions made, which could improve considerably the quality of the manuscript.
Here, below your comments, you can see the comments and changes made after revision in highlight.
- In the comfort assessment, authors stated the type of outsole were blinded, however, blindness might not be certified because it will be easy for some patients to feel their outsole rigidity. Moreover, shoe upper surfaces look a little bit different between rigid and semi-rigid in Fig.1.
In spite of shoe upper surfaces were a little bit different between rigid and semi-rigid outsole footwear (Figure 1) that could affect the blindness in some patients, comfort sensation becomes a subjective feeling in which several factors are influenced including appearance. This statement was added to the discussion section.
Lines 336 to 337.
- In description of shoe conditions, laces are not seen in Fig.1. Strictly speaking, laces and buckles are different for the ankle fixation and thus will affect the parameters the authors described. Type of shoes should be intimately described if there are some significant differences about parts.
All the shoes had buckles and thus we did not analyze shoes with laces, it has been corrected in the main text in the shoe condition part.
Lines 128 and 129.
- There seems to be big differences in foot type. Therefore, foot type variability might be correlated with the result. Sample size is limited but reference to the effective foot type for the rigid sole is helpful to the readers.
Foot position was stratified according to the foot posture index (FPI) into neutral, pronated, highly pronated, supinated and highly supinated. Quantitative values of FPI was changed by description of each foot type n (%) (Table 1). To show differences between foot type and the possible implications on spatio – temporal parameters during gait with the use of both different densities of rocker soles, Kruskal – Wallis test was used.
Table 1, line 219.
Statistical analyses, lines 201 and 202.
Table 3, line 293 and 294.
Results, line 320
- Typo error: P3 disfunction->dysfunction.
The typo error was corrected as the reviewer has noted.
Line 70
Reviewer 2 Report
Thank you for your contribution to footwear impact to gait. I have a number of major concerns about the design and interpretation of this study outlined below that should be address before publication should be considered:
While you have developed a study that investigates the difference between two footwear types, I'm not convinced we would have seen any different to that already known with rocker soled footwear. All of which have been extensively studies and no additional rational as to why it would be different with the foot with a previous history of ulceration. This has not been addressed in the manuscript and detracts greatly from the novelty.
As the authors point out, ulceration is both a mix of limb disease and the interface between the foot and the ground. While modification of the shoe mitigates the interface, what is missing from this research is the change in plantar pressure that the different gait parameters bring. This is the critical that is missing and allows the information to be usable. If we shorten (or lengthen) different components of gait, what happens to the plantar pressure. If you increase velocity, does this increase pressure? This has not been considered and is a major flaw in the design and interpretation of the data.
Statistical analysis - the authors have used data from both limbs. While they make the argument this is due to independence, there is no data provided to allow the reader to have confidence in this. Given the tight inclusion criteria, more data is required from left and right limbs to allow us to have confidence in the independence of that data. Particularly as one rational is the difference in FPI. The FPI-6 is a 24 point scale that generally (and not referenced) describes a foot posture but more recently avoids classifying into supinated, pronated and neutral. Based on the presentation of the data, the majority of people appear to have a similar foot posture. There is not enough detail of foot deformity or amputation to allow us to understand the differences between feet.
In addition, the authors are reporting median data (between timepoint) for within subject differences. The analysis does not allow for direction to be inferred based on the statistical analysis undertaken, just that there were differences between timepoints.
Limited discussion as to how the footwear was blinded to participants and randomised. Was their blinding in analysis?
I strongly disagree with the statement that "therapeutic footwear is the primary strategy for preventing foot disease and it's complication". Preventing foot disease is multifactorial, including tight HBA1C levels, weight control, vascular status, foot insoles, footwear etc all have leading roles. Placing the weight of prevention on footwear is medically unsound.
Authors have described where the pivot points are required on the footwear to minimise ulceration yet not described how this was customised based on the different foot types of people recruited in the trial. How can the reader be confident that the pivot point was at the right point if the rocker was based on foot size rather than an individualised modification?
Use of a comfort scale - how is this valid in a population with neuropathy?
Minor concerns:
Minor grammar and spelling throughout the manuscript was found
Statistical presentation requires editing as use of hyphens makes many of the results look like they are negative values.
Removal of non-standard abbreviations. These make the article extremely hard to read and therefore the reader spends more time trying to remember the abbreviation and less about what is being conveyed.
Author Response
Thank you for accepting the review of the current article and for the suggestions made, which could improve considerably the quality of the manuscript.
Here, below your comments, you can see the comments and changes made after revision in highlight.
- While you have developed a study that investigates the difference between two footwear types, I'm not convinced we would have seen any different to that already known with rocker soled footwear. All of which have been extensively studies and no additional rational as to why it would be different with the foot with a previous history of ulceration. This has not been addressed in the manuscript and detracts greatly from the novelty.
The rationale of the study was changed and explained accordingly in order to add novelty to the research.
Lines 64 to 75.
- As the authors point out, ulceration is both a mix of limb disease and the interface between the foot and the ground. While modification of the shoe mitigates the interface, what is missing from this research is the change in plantar pressure that the different gait parameters bring. This is the critical that is missing and allows the information to be usable. If we shorten (or lengthen) different components of gait, what happens to the plantar pressure. If you increase velocity, does this increase pressure? This has not been considered and is a major flaw in the design and interpretation of the data.
The aim of our study was not to analyze plantar pressure in relationship with spatio – temporal parameters during gait, the relationship between plantar pressure and changes in spatio-temporal parameters has been discussed in the text as a limitation.
Lines 312 to 318.
- Statistical analysis - the authors have used data from both limbs. While they make the argument this is due to independence, there is no data provided to allow the reader to have confidence in this. Given the tight inclusion criteria, more data is required from left and right limbs to allow us to have confidence in the independence of that data. Particularly as one rational is the difference in FPI. The FPI-6 is a 24 point scale that generally (and not referenced) describes a foot posture but more recently avoids classifying into supinated, pronated and neutral. Based on the presentation of the data, the majority of people appear to have a similar foot posture. There is not enough detail of foot deformity or amputation to allow us to understand the differences between feet.
According to the suggested recommendations, details of foot deformity and previous minor amputation have been added into (Table 3) to allow a better understanding in the differences between feet of each patient. In addition, Kruskall – Wallis test was used to verify if there any correlation between foot type and spatio – temporal and kinematic parameters.
Statistical analyses, lines 201 and 202.
Table 3, line 292 to 294.
Results, line 320
- In addition, the authors are reporting median data (between timepoint) for within subject differences. The analysis does not allow for direction to be inferred based on the statistical analysis undertaken, just that there were differences between timepoints.
We modified it along the result section accordingly as the reviewer has noted.
- Limited discussion as to how the footwear was blinded to participants and randomised. Was their blinding in analysis?
Blinding procedures were performed in two aspects during the study:
1) Patient were blinded from the type of outsole used during study trials in order to avoid bias in the interpretation of shoe comfort scale. The shoes looked the same but the hardness of the outsole was different (semi – rigid or rigid).
2) The investigator who evaluated data of the kinematic analysis. And extracted data from spatio – temporal parameters was blinded from the type of outsole used during trials.
It was discussed accordingly.
Lines 181 and 182.
Lines 336 to 337.
- I strongly disagree with the statement that "therapeutic footwear is the primary strategy for preventing foot disease and it's complication". Preventing foot disease is multifactorial, including tight HBA1C levels, weight control, vascular status, foot insoles, footwear etc all have leading roles. Placing the weight of prevention on footwear is medically unsound.
We completely agree with the revision, it has been changed into the Introduction section accordingly.
Lines 57 and 58.
- Authors have described where the pivot points are required on the footwear to minimise ulceration yet not described how this was customised based on the different foot types of people recruited in the trial. How can the reader be confident that the pivot point was at the right point if the rocker was based on foot size rather than an individualized modification?
To ensure the proper position of the pivot point in the rocker outsole, the principal investigator used a weight-bearing lateral X-ray of the shod feet as previously described.
Lines 136 and 137.
- Use of a comfort scale - how is this valid in a population with neuropathy?
To the best of our knowledge this is the first study that investigate the comfort of therapeutic footwear with a comfort scale in patients with neuropathy and previous DFU. The comfort of usage is something subjective and thus it might be the conjunction of many factors such as autonomous gait, faster gait speed or even the external appearance of the footwear.
Line 338.
Minor concerns:
- Minor grammar and spelling throughout the manuscript was found
The paper was sent to an edition service again to ensure the language is clear and free of errors. We attached the certificate of English editing.
- Statistical presentation requires editing as use of hyphens makes many of the results look like they are negative values.
Statistical presentation was edited for a better appearance, and now it avoids the results looking like negative values.
- Removal of non-standard abbreviations. These make the article extremely hard to read and therefore the reader spends more time trying to remember the abbreviation and less about what is being conveyed.
Non – standard abbreviations were removed from the main text for a quicker reading according to the recommendations.
Round 2
Reviewer 2 Report
Thank you for your revision and efforts to amend. I still have substantial concerns about the paper, many of which haven't been addressed during the revision.
I commend the authors for having it English edited however there are still grammar and edits required in the abstract and parts of the manuscript that have been included. I will leave these to the editor for correction/assessment.
In reference to your amendments.
While the authors have provided additions for novelty, ad I understand that pressure hasn't been studied within these results, it still stands, that gait changes pressure - if you are changing or impacting gait, you will change/impact pressure. This should be considered and discussed within the novelty and again in the discussion. This is a critical link that is missing throughout the manuscript. Particularly if you increase speed with footwear, what will happen to pressure, is it possible that by increasing or improving gait, you increase pressure and therefore create a new risk? or vis versa, is increasing potentially a benefit or potential risk reduction. The impact of these gait changes and discussion within the article is basic and may lead clinicians to over or underestimate the results from this study. This was a primary concern during last review. I note the authors have introduced a small edit to address this however I don't believe it has addressed this concern.
Ln 57: I note the change however this statement is still overstating the effect of footwear based on the supporting evidence as a main treatment. It is a component of HBA1C control, vascular sufficiency, monitoring, insoles, padding, orthoses, medical grade or non (depending on need and foot type). The reference states the guidelines that are cautiously advocating for therapeutic footwear, and only in some cases, for some people. This reference should be further considered.
Ln 69:Edits have included language that isn't person centred and would urge authors to consider patients or person with.... rather than DPN patients. For example: https://journals.sagepub.com/doi/epub/10.1177/0145721717735535
Ln 69: Consider changing the word "behave". it isn't a concept relating to gait.
Ln 118: The stratification of foot posture into groups is not based on the publication referenced. I am unsure where this stratification is from as it is commonly accepted that adult normal foot posture is between +1 to +7. (https://link.springer.com/article/10.1186/1757-1146-1-6). The reference from the developers of the FPI-6 advocates for typical range based on pathology and I would urge the authors to justify a different grouping system. This potentially can impact how people "over treat" or consider foot appearance. This also impacts your results as you have grouped people into non-standard groups. If you are to do this, stronger justification is required. You also have not provided any information on how you have grouped people where the left or right foot is a different group to the other.
Ln 181: No inclusion (as requested) of how the shoe types were blinded to the participants as requested. This has only been provided in the discussion but not how it was done.
Ln 201: Analysis is to understand or explore rather than show any difference. However based on my previous comment on grouping, this analysis needs justification or re-analysis. If it was grouped in this way, power analysis (or the limitation of) should be explored or addressed in the limitations. As this was added, this wasn't addressed in the last edit.
Results: while I still have concerns about the use of comfort with the population I note that the authors have introduced this as a novel scoring system and justified it accordingly. However no comfort results have been presented within the results section, instead discussed only.
Author Response
LETTER TO REVIEWER
Thank you for the quick answer to the review of the current manuscript version and for the suggestions made, we are sure that it could improve considerably the quality of the manuscript.
Here, below your comments, you can see the comments and changes made after revision in highlight.
- I commend the authors for having it English edited however there are still grammar and edits required in the abstract and parts of the manuscript that have been included. I will leave these to the editor for correction/assessment.
The manuscript has been edited in order to avoid grammar and edits mistakes. The certificate of English editing is attached.
- While the authors have provided additions for novelty, ad I understand that pressure hasn't been studied within these results, it still stands, that gait changes pressure - if you are changing or impacting gait, you will change/impact pressure. This should be considered and discussed within the novelty and again in the discussion. This is a critical link that is missing throughout the manuscript. Particularly if you increase speed with footwear, what will happen to pressure, is it possible that by increasing or improving gait, you increase pressure and therefore create a new risk? or vis versa, is increasing potentially a benefit or potential risk reduction. The impact of these gait changes and discussion within the article is basic and may lead clinicians to over or underestimate the results from this study. This was a primary concern during last review. I note the authors have introduced a small edit to address this however I don't believe it has addressed this concern.
This concern was highly explained in the introduction (lines 77 to 77) and discussion (lines 325 to 342) section based on previous evidence in the literature. We used to explore the relationship between spatiotemporal parameters and plantar pressure changes in healthy persons and patients with diabetes and foot complications.
Further, we believe, that the changes observed in our research should decrease plantar pressure, due to our patients suffer from basal altered gait patterns (decreasing step speed, step length and midstance phase) secondary to the implementation of a security mechanism to avoid falls, secondary to alterations on proprioception and limited ranges of motion.
A harder rocker sole could normalize this previous statement (that increase plantar pressure) and reduce forefoot plantar pressure and further risk of ulcer occurrence.
- Ln 57: I note the change however this statement is still overstating the effect of footwear based on the supporting evidence as a main treatment. It is a component of HBA1C control, vascular sufficiency, monitoring, insoles, padding, orthoses, medical grade or non (depending on need and foot type). The reference states the guidelines that are cautiously advocating for therapeutic footwear, and only in some cases, for some people. This reference should be further considered.
The statement in the introduction was changed accordingly, it was added that therapeutic footwear is part of the preventive treatments described in the International Guidelines in high risk population (risk 2 and 3 of the IWGDF), in addition HBA1C control, vascular sufficiency, self – monitoring, padding, orthoses and medical grade was added. Lines 57 to 59.
- Ln 69:Edits have included language that isn't person centred and would urge authors to consider patients or person with.... rather than DPN patients. For example: https://journals.sagepub.com/doi/epub/10.1177/0145721717735535
It was changed according the suggested recommendations along the text.
- Ln 69: Consider changing the word "behave". it isn't a concept relating to gait.
The word “behave” was removed from the main text and was changed by “affected” as the reviewer suggested. Line 78.
- Ln 118: The stratification of foot posture into groups is not based on the publication referenced. I am unsure where this stratification is from as it is commonly accepted that adult normal foot posture is between +1 to +7. (https://link.springer.com/article/10.1186/1757-1146-1-6). The reference from the developers of the FPI-6 advocates for typical range based on pathology and I would urge the authors to justify a different grouping system. This potentially can impact how people "over treat" or consider foot appearance. This also impacts your results as you have grouped people into non-standard groups. If you are to do this, stronger justification is required. You also have not provided any information on how you have grouped people where the left or right foot is a different group to the other.
The stratification of Foot Posture Index was changed accordingly into, neutral, pronated and supinated feet according to the suggestions. In addition, the reference was changed to the one provided by the reviewer (Redmond AC, Crane YZ, Menz HB. Normative values for the Foot Posture Index. Journal of Foot and Ankle Research 2008, 1:6 doi:10.1186/1757-1146-1-6). Lines 121 and 122.
The FPI-6 was explained accordingly along the test and the Kruskal Wallis test was analyzed again to explore differences between spatiotemporal parameters and foot type.
We consider that the combination of previous amputation and foot type allows an individual statistical treatment separated from each limb.
- Ln 181: No inclusion (as requested) of how the shoe types were blinded to the participants as requested. This has only been provided in the discussion but not how it was done.
It was explained to give a better understanding of the blinding procedure as follows:
“The therapeutic footwear trial was conducted in random, using a random number table in order to elect the hardness of the outsole, and thus the patients were blinded to the type of outsole used during gait to avoid bias in the interpretation of the shoe comfort due to both footwears had the same appearance on the upper surface”. Lines 173 to 176.
- Ln 201: Analysis is to understand or explore rather than show any difference. However based on my previous comment on grouping, this analysis needs justification or re-analysis. If it was grouped in this way, power analysis (or the limitation of) should be explored or addressed in the limitations. As this was added, this wasn't addressed in the last edit.
The phrase “show differences” was changed by “understand” and “explore” in the statistical analyses section accordingly. Lines 202 and 204.
In the other hand, the Foot Posture Index was stratified into neutral, pronated and supinated foot as the reviewer suggested. It is true, that the word highly overestimates the foot type and also it is not published previously.
- Results: while I still have concerns about the use of comfort with the population I note that the authors have introduced this as a novel scoring system and justified it accordingly. However no comfort results have been presented within the results section, instead discussed only.
The comfort results are showed as following:
“With respect to comfort, VAS scores were higher in the rigid outsole condition (89 [94 – 81] mm) than in the semirigid sole condition (62.5 [75 – 49.2] mm) (P < 0.001; Cohen d = 1.38)” Lines 291 to 293.
The novelty of the VAS score system was marked in the main text (lines 170 to 172) and discussed as a limitation.
We did not perform any graphical analyses of the results due it was not the primary outcome.